# Colloidal Crystal Films with Narrow Reflection Bands by Hot-Pressing of Polymer-Grafted Silica Particles

**DOI:** 10.3390/polym14235157

**Published:** 2022-11-27

**Authors:** Sawa Matsuura, Mami Obara, Naoto Iwata, Seiichi Furumi

**Affiliations:** Department of Chemistry, Graduate School of Science, Tokyo University of Science, 1-3 Kagurazaka, Shinjuku, Tokyo 162-8601, Japan

**Keywords:** colloidal crystal, surface-initiated atom transfer radical polymerization, Bragg reflection, reflection color, silica particles

## Abstract

Previous reports have shown that colloidal crystal (CC) films with visible Bragg reflection characteristics can be fabricated by the surface modification of monodisperse silica particles (SiPs) with poly(methyl methacrylate) (PMMA) chains, followed by hot-pressing at 150 °C. However, the reflection bands of the CC films were very broad due to their relative disordering of SiPs. In this report, we attempted to fabricate the CC films using SiPs surface-modified with poly(*n*-octyl acrylate) (POA) chains by hot-pressing. When the cast films of POA-grafted SiPs were prepared by hot-pressing at 100 °C, the reflection bands were narrow rather than those of CC films of PMMA-grafted SiPs. This can be ascribed to easy disentanglement of POA chains during the hot-pressing process, thereby enabling the formation of well-ordered CC structures. Moreover, the reflection colors of CC films could be easily tuned by controlling the molecular weight of POA chains grafted on the SiP surface.

## 1. Introduction

Colloidal crystals (CCs) are highly ordered three-dimensional (3D) periodic architectures of monodisperse colloidal particles, which serve as one of the 3D photonic crystals [1,2,3,4,5]. Such CC systems allow the emergence of photonic band gap(s), wherein photons are strongly prohibited within the specific region(s) of electromagnetic waves determined by the periodic modulation in the refractive index of materials [6,7,8]. Therefore, the CCs can selectively reflect the light of a specific wavelength according to Bragg’s law. The reflection wavelength (*λ*) of CCs is numerically expressed by the following equation [3,9,10].
(1)λ=2ndsinθ
where *n* denotes the effective refractive index, *d* is the lattice space of the CC and *θ* is the incident angle of light from the horizontal plane, that is, the Bragg angle. Generally, the Bragg reflection of CCs stems from the light reflection from the (111) lattice plane of the face-centered cubic lattice of colloidal particles. Thus, Equation (1) can be rearranged to the following equation:(2)$$\lambda =\frac{2na}{\sqrt{3}}\sin\theta$$
where *a* corresponds to the lattice constant of colloidal particles. This reflection coloring phenomenon is permanently preserved if the CC structures are not destroyed. Therefore, uniform CC structures are technologically available as durable colorful pigments.

So far, the CCs have been widely studied using monodisperse colloidal particles, such as silica or polystyrene as their building blocks [3,11,12,13,14]. This is because they can be obtained as the commercial products from a large number of companies. Previously, Ohno and colleagues established an intriguing strategy to synthesize core-shell type hybrid colloidal particles consisting of silica particles as the core and poly(methyl methacrylate) (PMMA) layer as the shell. Since these hybrid colloidal particles, abbreviated as SiP-PMMA hereinafter, were synthesized by the surface-initiated atom transfer radical polymerization (SI-ATRP), the outermost surface of silica particles could be chemically modified by densely-grafted PMMA chains with their uniform lengths [15,16]. In other words, the PMMA shell thickness was controllable by the reaction condition of the SI-ATRP. As a result, the CCs were obtained by dispersing these hybrid particles of SiP-PMMA in mixed organic solvents, such as 1,2-dichloroethane, chlorobenzene or 1,2-dichlorobenzene [17,18]. Moreover, Ohno and his co-worker recently explored another strategy to fabricate CCs in a polymeric matrix [19]. When the CC films were prepared by drop-casting the SiP-PMMA suspensions on substrates, followed by hot-pressing at 150 °C, visible Bragg reflection emerged on the CC films. In this way, burdensome handling of SiP-PMMA is required for the fabrication of CC films, because the temperature of hot-pressing is relatively high. In addition, the grafted PMMA chains on the surface of SiPs play the roles of steric barriers to prevent the aggregation of SiPs as well as a polymeric matrix in which SiPs are arranged in the periodical manner to form the CC structures. However, the reflection bands of CC films comprising SiP-PMMA were very broad probably due to the disordered CC structures of SiPs. According to their precedent, it would seem that the addition of PMMA-grafted carbon black nanoparticles with a diameter of ~15 nm into SiP-PMMAs is indispensable to the spectral narrowing in the reflection bands of CC films due to the reduction of light scattering by carbon black nanoparticles.

In this study, we attempted to fabricate CC films exhibiting Bragg reflection colors with narrow reflection bands in the absence of carbon black nanoparticles. For this purpose, the outermost surface of SiPs was modified with poly(*n*-octyl acrylate) (POA) chains through the SI-ATRP. Such POA-grafted SiPs are herein abbreviated as SiP-POAs. Subsequently, the CC films were prepared by drop-casting the SiP-POA suspension, followed by hot-pressing at 100 °C. The Bragg reflection colors could be observed for the CC films of SiP-POA to be the same as those of SiP-PMMA. Importantly, the reflection bands of CC films with SiP-POA were adequately narrow even though carbon black nanoparticles were not doped in the CC films. Taking account of the empirical fact that bulk POA shows the glass transition temperature at −62 °C [20], which is much lower than that of bulk PMMA, it can be considered that the POA chains on the surface of SiPs are more likely to be disentangled during the hot-pressing process. Therefore, well-defined CCs of SiP-POAs could be prepared only by hot-pressing without the aid of carbon black nanoparticles, which is evident from the narrow reflection bands. The present report indicates that SiP-POA might be suitable for the fabrication of CC films with visible reflection features only by the hot-pressing process.

## 2. Experimental Section

An aqueous suspension of monodisperse SiPs (Seahostar KE-W10) was purchased from Nippon Shokubai Co., Ltd. (Osaka, Japan). Before experiments, microscopic observation of the SiPs using scanning electron microscopy (JSM-7800F Prime, JEOL, Tokyo, Japan) confirmed that the average diameter and its coefficient of variation of SiPs are ~120 nm and 7.6%, respectively (Appendix A). (2-Bromo-2-methyl)propionyloxyhexyltriethoxysilane (BHE), which was used as a silane coupling agent possessing the ATRP initiating group, was synthesized according to the procedure reported by Ohno and co-workers [15]. *n*-Octyl acrylate (OA) and ethyl 2-bromoisobutyrate (EBIB) were obtained from Tokyo Chemical Industry Co., Ltd. (Tokyo, Japan). Ammonia solution, copper(I) bromide, copper(II) bromide and tris[2-(dimethylamino)ethyl]amine (Me_6_TREN) were obtained from FUJIFILM Wako Pure Chemical Corporation (Osaka, Japan). Tributyltin hydride was obtained from Sigma-Aldrich Japan (Tokyo, Japan). OA as a monomer was passed through a column filled with activated basic alumina to remove the polymerization inhibitor before use. All the other reagents were used as received from commercial products.

A series of SiP-POAs with different molecular weights of POA chains were prepared here by the SI-ATRP using SiPs with a diameter of ~120 nm, as depicted in Figure 1.

In the first step, the outermost hydroxy groups of SiP were modified with the ATRP initiators through the silylation with BHE according to the procedure by Ohno and colleagues [15]. The commercially available SiP suspension in water was solvent-exchanged to ethanol by three rounds of centrifugation at 1.0 × 10^4^ rpm for 40 min and dispersion in fresh ethanol. At this time, the concentration of ethanolic SiP suspension (25.0 g) was adjusted to ~10 wt%. Then, BHE (2.00 g, 4.80 mmol) and 28% aqueous solution of ammonia water (15.4 mL) as an alkaline catalyst were added dropwise to this suspension and stirred at 40 °C for 18 h. The SiPs surface-modified with the ATRP initiators, abbreviated as SiP-Br, were collected by centrifugation. After the supernatant was removed, SiP-Br was dispersed in fresh ethanol. This process was repeated three times to completely remove an excess of initiators and ammonia.

In this study, three kinds of SiP-POAs with different molecular weights of POA chains were synthesized by changing the reaction condition of the SI-ATRP, as listed in Table 1. The number in the sample code represents the molecular weight of POA, as will be shown in Table 2.

For example, SiP-POA109k was synthesized by the following procedure. First, SiP-Br was dispersed in OA (30.0 g, 155 mmol) at a concentration of 1.3 wt%. To this suspension, EBIB (1.97 mg, 10.2 μmol) as an initiator of the ATRP, Me_6_TREN (28.0 mg, 124 μmol) as a ligand and Cu(II)Br (6.80 mg, 30.4 μmol) as a catalyst were added and bubbled with nitrogen for 30 min to purge oxygen. Then Cu(I)Br (13.6 mg, 94.8 μmol) as a catalyst was quickly added to the suspension and bubbled with nitrogen for an additional 10 min. After that, the suspension was heated at 45 °C and stirred for 2.5 h. The SI-ATRP was quenched by adding tributyltin hydride (43.2 mg, 148 μmol) to the mixture and stirring for 1 h. Subsequently, the reaction mixture was cooled down to room temperature, and bubbled with air to ensure the termination of ATRP. Successively, this reaction mixture was diluted with tetrahydrofuran (THF) and centrifuged at 1.0 × 10^4^ rpm for 40 min to collect SiP-POA. The supernatant was separated as it contained unbonded POA chains, which were initiated from EBIB. This cycle of centrifugation and redispersion in THF was repeated three times. Finally, the sediment was dispersed into THF and poured into methanol, and dried to yield the purified SiP-POA. The solution of POA initiated from EBIB was passed through an activated alumina column and purified by reprecipitation from THF to methanol and dried.

The number average molecular weight (*M*_n_) and its distribution (*M*_w_/*M*_n_) of POA chains which were initiated from EBIB grown at the same time as the grafted chains were determined by size-exclusion chromatography (SEC, HLC-8220GPC, TOSOH, Tokyo, Japan) measurements assuming that the values of *M*_n_ and *M*_w_/*M*_n_ are nearly equal to those of grafted on particles [15]. In the SEC measurements, THF was used as the eluent. Polystyrene standards were used to calibrate the *M*_n_ and *M*_w_/*M*_n_ values. It should be noted that both *M*_n_ and *M*_w_/*M*_n_ of these polymers are almost equal to those grafted on SiPs, according to the previous report [15]. The weight fractions of POA chains in SiP-POAs were determined by measuring the weight loss of SiP-POAs using a thermogravimetric analysis (TGA) system (TGA 2010SA, NETZSCH, Selb, Germany). The TGA measurements were conducted in the temperature range between 25 °C and 450 °C at the heating rate of 10 °C/min under a nitrogen atmosphere.

The CC films of three kinds of SiP-POAs with different molecular weights of POA chains were prepared by solvent-casting of the SiP-POA suspensions in toluene, followed by hot-pressing at 100 °C. Typically, SiP-POA was dispersed in toluene at a concentration of ~20 wt%. This suspension was dropped in a polytetrafluoroethylene mold and slowly dried over 2 days at room temperature under an atmosphere of toluene vapor. After that, the dried SiP-POA was hot-pressed by hand at 100 °C for 1 h in order to offer a solid-state CC film of SiP-POA (Appendix A). Reflection spectra were measured using a spectrometer (USB2000+, Ocean Optics, FL, USA) combined with a white light source (HL-2000, Ocean Optics). The incident light was irradiated from the vertical direction of the film surface. The Commission Internationale de l’Éclairage (CIE) color coordinate values were obtained using a software (SpectraSuite, Ocean Optics) for the spectrometer.

## 3. Results and Discussion

### 3.1. Synthesis and Characterization of SiP-POA

The *M*_n_, *M*_w_/*M*_n_ and graft density (*σ*) of SiP-POAs with different molecular weights of POA chains are compiled in Table 2. The *M*_n_ and *M*_w_/*M*_n_ values for the grafted chains are assumed to be the same as the POA chains initiated from EBIB according to the previous report [15].

Since the *M*_w_/*M*_n_ values of POAs initiated from EBIB were estimated to be ~1.13, the surface of SiPs could be modified with POA chains which have uniform length. The *M*_n_ values of POA chains were easily controlled in the range from 1.09 × 10^5^ to 2.78 × 10^5^ by changing the polymerization conditions. Such fine control of *M*_n_ with narrow *M*_w_/*M*_n_ can be achieved by adopting the SI-ATRP since it is regarded as a kind of controlled radical polymerization. The *σ* value of SiP-POAs was calculated by the following equation [21].
(3)$$\sigma =\frac{1}{4\pi r^{2}}\left[\left(\frac{4}{3}\pi r^{3}\rho \right)\frac{\varphi_{w}}{100-\varphi_{w}}\right]\frac{N_{A}}{M_{n}}$$
where *r* stands for the average radius of SiPs, *ρ* is the density of SiPs, *φ*_w_ is the weight fraction of POA and *N*_A_ is the Avogadro constant. In this study, we adopted the *ρ* value of 2.20 g/cm^3^ from the data sheet of Nippon Shokubai Co., Ltd. It should be noted that the *φ*_w_ can be estimated from the TGA measurements, as described below.

TGA curves of SiP, SiP-Br, bulk POA and SiP-POAs with different *M*_n_ values are shown in Figure 2. The thermal decomposition of SiP hardly occurred after heating up to 450 °C (Figure 2A, black curve) because silica is a heat-resistant inorganic material. The weight loss of SiP at 124 °C can be ascribed to water adsorbed on the surface of SiP. With regard to the result of SiP-Br, the weight loss reached 8.06% (Figure 2A, gray curve), which is larger than that of SiP. This result infers that the SiP surface is modified with initiators for the SI-ATRP, corresponding to BHE.

On the contrary, three kinds of SiP-POAs exhibited relatively large values of weight loss as compared to that of SiP-Br (Figure 2B). These results indicate that the surface of SiPs is modified with POA chains with different *M*_n_ values. Considering the fact that bulk POA completely decomposes by heating at 427 °C (Figure 2A, dashed gray curve), the weight loss of SiP-POA at approximately 435 °C can be ascribed to the *φ*_w_ value of SiP-POA. In the case of SiP-POA109k, the weight loss was found to be 76.9% by TGA measurement. Thus, the *σ* value was calculated to be 0.81 chains/nm^2^. The *σ* values of SiP-POA191k and SiP-POA278k were also calculated by the same procedure. It is well known that the polymer chains grafted on particle surfaces take well-extended conformations when *σ* value exceeds 0.1 chains/nm^2^ [22,23]. Indeed, as given in Table 2, the *σ* values of SiP-POAs were in the range of 0.45–0.81 chains/nm^2^, implying that the surface of SiPs can be modified with POA chains with high graft densities.

### 3.2. Fabrication and Reflection Properties of SiP-POA Films

As the toluene suspensions of SiP-POAs were cast and hot-pressed according to the procedure mentioned in the Experimental Section, the solid-state CC films of SiP-POAs could be obtained without any covalent crosslinking. The slow evaporation of suspensions of SiP-POAs in toluene offered the CC films of SiP-POA with a thickness of ~100 μm. Some of these films were piled together and hot-pressed at 100 °C to obtain a SiP-POA film with a thickness of ~300 μm. By considering the fact that the glass transition temperature of bulk POA is low when compared to that of bulk PMMA [24], the hot-pressing of SiP-POA at 100 °C ensured the formation of CC films with reflection colors. The resultant SiP-POA films were obtained as free-standing solid-state films. This can be ascribed to the entanglement of POA chains grafted on the surface of SiPs, which is reasonable since the entanglement molecular weight of POA has been reported as 1.52 × 10^4^ [25].

Interestingly, we found that the SiP-POA films show reflection colors depending on the molecular weight of POA chains. The reflection spectra of CC films of SiP-POAs are shown in Figure 3A. The reflection band was primarily observed at 462 nm for the CC film of SiP-POA109k, 512 nm for SiP-POA191k and 602 nm for SiP-POA278k. Our preliminary experiments revealed that the reflection band cannot be observed in the visible wavelength range when the CC film is prepared by hot-pressing of SiP-POA with a molecular weight of 0.59 × 10^5^ due to short POA chains. Therefore, the molecular weight of POA on the SiP might be controlled over ~1.00 × 10^5^ to prepare the CC film with visible reflection. Moreover, we found that the reflection colors of CC films of SiP-POA109k and SiP-POA191k can be easily recognized by the naked eye as blue and green, respectively (Figure 3A, Insets *a* and *b*). On the other hand, the CC film of SiP-POA278k showed muted red as the reflection color probably due to a slight disordering of the internal colloidal particles by relatively long POA chains in the exterior shell (Figure 3A, Inset *c*). As expected, the reflection properties of CC films showed strong angle dependency. The reflection wavelength of CC films was blue-shifted when they were viewed from an oblique angle (Appendix A). This is reasonable because the reflection wavelength of the CC is dependent on the Bragg angle, as shown in Equation (1).

In order to obtain further insight into reflection properties, we numerically analyzed the CIE color coordinates values for three kinds of CC films fabricated from SiP-POA109k, SiP-POA191k and SiP-POA278k. The reflection spectra of the CC films were converted to the CIE color coordinate values to create the plots on the CIE diagram. As a result, the (*x*, *y*) coordinate values of the three CC films were found to be (0.43, 0.39) for SiP-POA109k, (0.44, 0.42) for SiP-POA191k and (0.46, 0.41) for SiP-POA278k (Appendix A). From these results, we concluded that the CC films can be potentially applied to color materials by hot-pressing SiP-POAs.

The mechanism for the emergence of different Bragg reflection colors on CC films of SiP-POAs can be explained as follows. The interparticle distance between SiP-POAs, that is, the lattice constant, was increased accompanied by the increase of the molecular weight of POA chains grafted on the surface of SiPs. Figure 3B shows the SEM images of CC films of SiP-POAs. As the molecular weight of POA chains increased, the neighboring distance between SiP-POAs also extended in a synchronous way. The interparticle distance for each CC film calculated from these SEM images was 190 nm for the CC films of SiP-POA109k, 200 nm for SiP-POA191k and 213 nm for SiP-POA278k. This is reasonable because the POA chain length becomes longer with the increase of its molecular weight. In this way, the maximum reflection wavelength was continuously red-shifted with the increase of molecular weight of POA chains caused by the enlargement of the lattice constant in accordance with Bragg’s law, as given in Equation (2). This result indicated that the Bragg reflection colors of CC films of SiP-POAs can be easily tuned by controlling the molecular weight of POA chains grafted on SiP surfaces through the SI-ATRP.

The POA chains on the surface of SiPs are assumed to be taking highly-stretched conformation. Figure 4 shows the dependence of the maximum reflection wavelengths of CC films of SiP-POAs on their degrees of polymerization of surface-grafted POA chains (*m*), as depicted in Figure 1. The reflection wavelength was proportionally red-shifted upon increasing the *m* value. This relationship between the Bragg reflection wavelength and molecular weight of POA chains can be explained as follows. The colloidal particles with surface-grafted polymer chains tend to form face-centered cubic lattices. According to the previous reports by Bockstaller and co-workers [23,26], the relationship between the lattice constant of CC structure, corresponding to *a* in Equation (2), and the degree of polymerization of surface-grafted polymer chains, corresponding to *m* in Figure 1, can be written as follows.
(4)$$a\sim m^{x}$$
where *x* is the scaling factor which depends on the graft density of polymer chains. The grafted polymer chains form highly-extended conformation to show a high *x* value, which is in the range from 0.8 to 1.0 when the graft density is large enough. Taking account of the relation that the Bragg reflection wavelength is proportional to the lattice constant of CCs as notated in Equation (2), the following relationship can be derived.
(5)$$\lambda \sim m^{x}$$

Since the Bragg reflection wavelengths of CCs from SiP-POAs are linearly dependent on the molecular weights of POA, it can be assumed that the POA chains grafted on SiPs form the highly-stretched conformation.

Previously, Ohno and his co-worker reported the reflection properties of CC films of SiP-PMMAs [19]. At this time, the reflection bands of CC films were very broad. In this study, we focused on the full width at half maximum (FWHM) in the reflection band in order to compare quantitatively the reflection properties of CC films of SiP-PMMAs and SiP-POAs. The FWHM values are shown in Table 3. From the previously reported reflection spectra of CC films with SiP-PMMAs, the FWHM values were estimated to be ~67 nm, ~160 nm and ~190 nm for three kinds of CC films with blue, green and red reflection colors, respectively [19]. Such broadening in Bragg reflection bands can be ascribed to the disordered CC structures of SiP-PMMAs. On the other hand, these FWHM values could be reduced to 30–60 nm by doping a tiny amount of PMMA-grafted carbon black nanoparticles into the CC films of SiP-PMMAs (Table 3). It should be noted that the addition of carbon black nanoparticles is not relevant for the formation of well-defined CC structures because the diameter of carbon black nanoparticles (15 nm) is much smaller than that of silica microparticles (120 nm). The spectral narrowing in the reflection bands arose from the reduction of light scattering by the light absorption of the carbon black nanoparticles in a wide visible wavelength range [19].

Such difficulty in fabricating uniform CC structures by utilizing SiP-PMMAs can be attributed to the entanglement of PMMA chains on the surface of SiPs during the hot-pressing process. Since the glass transition temperature of bulk PMMA has been reported as 100 °C [24], it can be assumed that the PMMA chains were still entangled at 150 °C, which inhibited the rearrangement of SiPs to provide CC structures. In contrast to SiP-PMMAs, the CC films of SiP-POAs exhibited narrow reflection bands even in the absence of carbon black nanoparticles, suggesting the formation of well-defined CC structures. Actually, the FWHM value in the reflection band was ~13 nm for the blue reflective CC film of SiP-POA109k, ~17 nm for the green reflective CC film of SiP-POA191k and ~23 nm for the red reflective CC film of SiP-POA278k. These values are approximately one-tenth of that of the CC films with SiP-PMMAs in the absence of carbon black nanoparticles (Table 3). It can be assumed that POA chains are disentangled during the hot-pressing process because the glass transition temperature of bulk POA is known to be as low as −62 °C [20]. As a result, SiPs are more likely to assemble well-defined CC structures because SiPs are dispersed in the melt of unentangled POA chains. These CC structures can be preserved by the entanglement of POA chains on the surface of SiPs to show stable Bragg reflection colors at room temperature. In this way, the grafting polymerization of POA chains on the SiP surface facilitated not only handling in the fabrication of CC films such as hot-pressing at relatively low temperature, but also spectral narrowing in Bragg reflection bands without the addition of carbon black nanoparticles by well-defined CC structures. From these results, we concluded that the SiP-POAs might be adequate to the preparation of uniform CC films with Bragg reflection colors. We are currently studying how to evaluate the uniformity of CC structures of SiP-POAs by scanning electron microscopy, which will be described in our forthcoming reports.

## 4. Conclusions

In this report, we successfully fabricated CC films with narrow Bragg reflection bands only by hot-pressing SiPs surface-modified with POA chains (SiP-POAs). The narrowness of the reflection bands can be explained by the formation of well-defined CC structures enabled by the disentanglement of POA chains during the hot-pressing process at 100 °C, stemming from the low glass transition temperature of bulk POA. Furthermore, the Bragg reflection colors on CC films of SiP-POAs were easily tuned by controlling the molecular weight of POA chains grafted on the SiP surface because of the linear relationship between the Bragg reflection wavelength of CC films and the degree of polymerization of POA chains by their well-extended conformations. We propose that SiP-POA is suitable for the fabrication of uniform CC films with visible reflection, which are desired for applications of next-generation lightweight photonic devices.

## Figures and Tables

**Figure 1 polymers-14-05157-f001:**
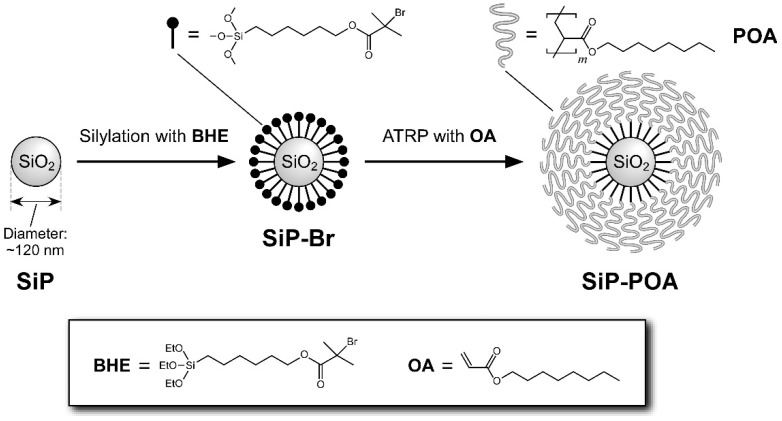
Schematic representation of synthesis of SiP-POA by the SI-ATRP.

**Figure 2 polymers-14-05157-f002:**
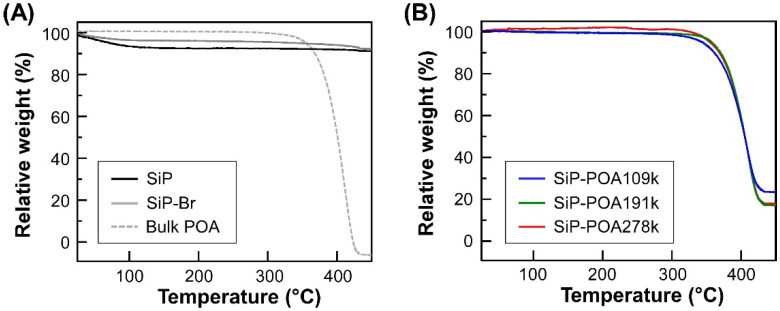
TGA curves for SiP, SiP-Br, bulk POA (**A**) and three kinds of SiP-POAs with different molecular weights of POA chains (**B**).

**Figure 3 polymers-14-05157-f003:**
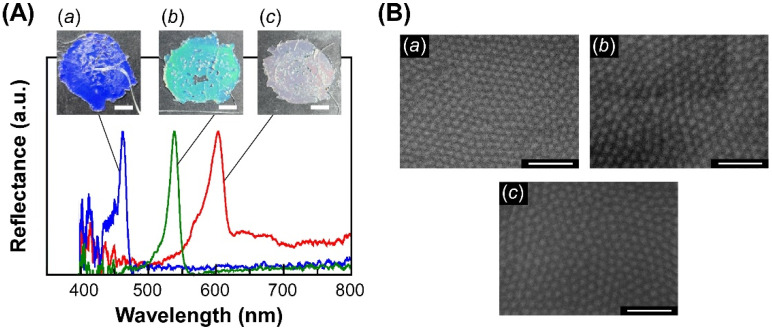
Reflection characteristics of CC films of three kinds of SiP-POAs with different molecular weights of POA chains. (**A**) Reflection spectra of CC films of SiP-POA109k (*a*), SiP-POA191k (*b*) and SiP-POA278k (*c*) fabricated by hot-pressing at 100 °C. Insets represent the reflection images of CC films, and the white scale bars signify 5 mm. (**B**) SEM images of the CC film surfaces of SiP-POA109k (*a*), SiP-POA191k (*b*) and SiP-POA278k (*c*) observed from the top view. White scale bars signify 1 µm.

**Figure 4 polymers-14-05157-f004:**
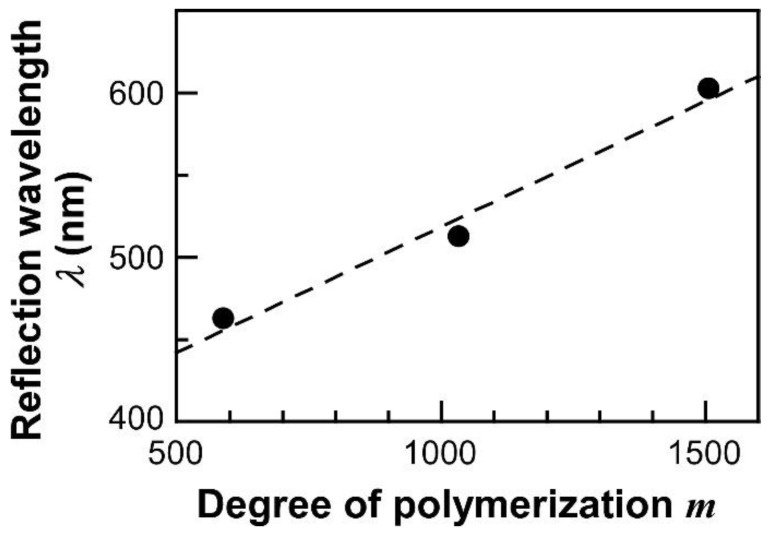
The relationship between the degrees of polymerization (*m*) of POA chains on the surface of SiPs and the maximum wavelengths of reflection bands (*λ*) of the CC films. The dashed line represents the linear regression line for the plots.

**Table 1 polymers-14-05157-t001:** Reaction conditions of the SI-ATRP for the synthesis of SiP-POAs.

Sample Code	EBIB (mg)	Me_6_TREN (mg)	Cu(I)Br (mg)	Cu(II)Br (mg)	Polymerization Time (h)
SiP-POA109k	1.97	28.0	13.6	6.80	2.5
SiP-POA191k	1.97	28.0	13.1	7.10	5.3
SiP-POA278k	1.97	28.0	16.8	6.90	3.5

**Table 2 polymers-14-05157-t002:** Characterization of SiP-POAs.

Sample Code	Weight Fraction of POA *φ*_w_ (%) ^(1)^	*M*_n_ (×10^5^) ^(2)^	*M*_w_/*M*_n_ ^(2)^	Graft Density *σ* (Chains/nm^2^)
SiP-POA109k	76.9	1.09	1.16	0.81
SiP-POA191k	83.2	1.91	1.14	0.69
SiP-POA278k	82.5	2.78	1.10	0.45

^(1)^ Determined by TGA measurements. ^(2)^ Determined by SEC measurements.

**Table 3 polymers-14-05157-t003:** Quantitative comparison of the FWHM values in reflection bands of CC films fabricated with SiP-PMMAs and SiP-POAs.

Sample	Reflection Colors of CC Films		
	Blue	Green	Red
SiP-PMMAs (w/o carbon black nanoparticles) ^(1)^	~67 nm	~160 nm	~190 nm
SiP-PMMAs (w/carbon black nanoparticles) ^(1)^	~30 nm	~56 nm	~63 nm
SiP-POAs (w/o carbon black nanoparticles) ^(2)^	~13 nm	~17 nm	~23 nm

^(1)^ The FWMH values were calculated from the reflection spectra reported in Ref. [19]. ^(2)^ The FWMH values were calculated from the reflection spectra in Figure 3A.

## Data Availability

Data is contained within the article.

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
