# Peer review of "Colloidal Crystal Films with Narrow Reflection Bands by Hot-Pressing of Polymer-Grafted Silica Particles"

_polymers, 2022, doi:10.3390/polym14235157_

Round 1

Reviewer 1 Report (Previous Reviewer 1)

I am now glad to recommend publication of the manuscript in its present form.

Reviewer 2 Report (Previous Reviewer 2)

The reviewer does not have further comments. 

This manuscript is a resubmission of an earlier submission. The following is a list of the peer review reports and author responses from that submission.

Round 1

Reviewer 1 Report

By discussing the results in terms of the quantitative FWHM of the reflection spectra of the CC films (revised manuscript), instead of describing the results in terms of the qualitative "vivid colors" (original manuscript), the authors fully answered the main concern I had with the original manuscript. In my opinion that is a major improvement of the manuscript. Together with the response to my additional comments I can now recommend publication of the manuscript after considering one minor comment/suggestion.

Minor comment/suggestion:
In line 255-256 it is written: "Since the Bragg reflection wavelengths of CCs from SiP-POAs are proportional to the molecular weights of POA, it can be assumed...". As far as I understand it, this does not agree with fig. 3B (the extrapolated curve is far from going through the origin), and "are proportional to" should be replaced with "are linearly dependent on", or some similar formulation. Or do I misunderstand something?

Author Response

Manuscript ID: polymers-1895318

Manuscript Title:

Colloidal Crystal Films with Narrow Reflection Bands by Hot-Pressing of Polymer-Grafted Silica Particles

(Previous MS title: Colloidal Crystal Formation of Polymer-Grafted Silica Particles Previous Manuscript ID: polymers-1810530)

Authors:

Sawa Matsuura, Mami Obara, Naoto Iwata* and Seiichi Furumi*

Point-by-point responses. Our responses and opinions are shown in BLUE.

Reviewer #1

============================================================

To Reviewer #1

Thank you very much for your kind review to our report. Our manuscript has been revised according to your comments.

============================================================

Comment #1

By discussing the results in terms of the quantitative FWHM of the reflection spectra of the CC films (revised manuscript), instead of describing the results in terms of the qualitative "vivid colors" (original manuscript), the authors fully answered the main concern I had with the original manuscript. In my opinion that is a major improvement of the manuscript. Together with the response to my additional comments I can now recommend publication of the manuscript after considering one minor comment/suggestion.

Response#1

We appreciate this highly encouraging comment for our revised manuscript.

Comment #2

Minor comment/suggestion:

In line 255-256 it is written: "Since the Bragg reflection wavelengths of CCs from SiP-POAs are proportional to the molecular weights of POA, it can be assumed...". As far as I understand it, this does not agree with fig. 3B (the extrapolated curve is far from going through the origin), and "are proportional to" should be replaced with "are linearly dependent on", or some similar formulation. Or do I misunderstand something?

Response#2

Based on your suggestion, we have revised the sentences in Lines 262-264, Page 7 of this revised manuscript.

Since the Bragg reflection wavelengths of CCs from SiP-POAs are linearly dependent on the molecular weights of POA, it can be assumed that the POA chains grafted on SiPs form the highly-stretched conformation.

Reviewer 2 Report

The reviewer does not have further comments.

Author Response

Manuscript ID: polymers-1895318

Manuscript Title:

Colloidal Crystal Films with Narrow Reflection Bands by Hot-Pressing of Polymer-Grafted Silica Particles

(Previous MS title: Colloidal Crystal Formation of Polymer-Grafted Silica Particles Previous Manuscript ID: polymers-1810530)

Authors:

Sawa Matsuura, Mami Obara, Naoto Iwata* and Seiichi Furumi*

Point-by-point responses. Our responses and opinions are shown in BLUE.

Reviewer #2

============================================================

To Reviewer #2

Thank you very much for your kind review and highly encouraging comments to our manuscript.

============================================================

Comment #1

The reviewer does not have further comments.

Response#1

We appreciate for taking the time to review our manuscript.

Reviewer 3 Report

Please see attached review.

Author Response

Manuscript ID: polymers-1895318

Manuscript Title:

Colloidal Crystal Films with Narrow Reflection Bands by Hot-Pressing of Polymer-Grafted Silica Particles

(Previous MS title: Colloidal Crystal Formation of Polymer-Grafted Silica Particles Previous Manuscript ID: polymers-1810530)

Authors:

Sawa Matsuura, Mami Obara, Naoto Iwata* and Seiichi Furumi*

Point-by-point responses. Our responses and opinions are shown in BLUE.

============================================================

To Reviewer #3

Thank you very much for your kind review and highly encouraging comments to our manuscript.

============================================================

Comment #1

The manuscript reviewed is the resubmitted version of “Colloidal Crystal Films with Narrow Reflection Bands by Hot Pressing of Polymer-Grafted Silica Particles”. The authors have also supplied their response to the original review, and the detailed responses and additions/changes to the manuscript cover most of the issues raised in the original review.

The major concern raised by Reviewer #2 regarding structural characterization of the hot-pressed films (Comment #2 in the response document) has not been adequately addressed and remains a major flaw in the manuscript. The authors argue the different spectral characteristics of the three different molecular weight SiP-POA films is due to differences in the inter-particle spacing. While this is a reasonable hypothesis, no direct evidence is given to support this. In the response document the authors show three SEM images of cast films; on of SiP and two of SiP-POA191k. Arguably, images (B) and (C) demonstrate the silica particles space further apart in the film due to the POA layer. The authors must include images such as these in the manuscript, and include images with the two other SiP-POA samples. The response from the authors to Reviewer #2 is not adequate, as careful casting of the three different SiP-POA samples should provide evidence of molecular weight dependent spacing if it exists. Until evidence is presented to support the hypothesis that the particles are differently spaced with different molecular weight POA coating the manuscript should not be accepted for publication.

The are a number of concerns that should be addressed prior to acceptance and publication:

Comment #1

As stated above, the critical flaw is the absence of evidence related to the structure of the silica particles in the thin films.

Response#1

In the previous work by Ohno and Mizuta (Ref. 19), SEM observation was conducted for CC films of SiP-PMMAs with different molecular weights of PMMA chains. This report has shown that the interparticle distance between SiP-PMMAs increases as the molecular weight of PMMA chains grafted on the surface of SiPs become higher to emerge different Bragg reflection peak wavelengths. Therefore, we consider that this mechanism can be also applicable to the CC films of SiP-POAs prepared in this study.

In response to the suggestion raised by the reviewer, we would like to point out that the molecular weight-dependent spacing of particles can NOT be observed for the CC film fabricated by careful casting. Because the CC films were not prepared by solvent-casting from a dilute suspension of SiP-POAs but by solvent-casting followed by hot-pressing, leading to the difference in interparticle distance. Thus, we believe that the structural characterization of CC films by SEM or ultra-small angle X-ray scattering (USAXS) is indispensable to understanding the molecular weight dependence of Bragg reflection colors of CC films. As noted in the previous peer-review response, we are currently trying to find an effective procedure for the SEM observations of CC films from SiP-POAs and will be reported in the future.

Comment #2

Line 82 – the SEM images of the commercial silica particles should be included in the Supplementary Material file at least.

Response#2

Based on your suggestion, the SEM image of the commercial silica particles has been added as Figure S1 in the Supplementary Materials.

Comment #3

Line 105 – I believe “corrected” should be “collected”.

Response#3

Thank you for pointing out our typo. We have revised the sentences in Lines 105-106, Page 3 of this revised manuscript as follows.

The SiP surface-modified with the ATRP initiators, abbreviated as SiP-Br, were collected by centrifugation.

Comment #4

Line 114 – Are the SiP-Br particles redispersed in ethanol for the ATRP synthesis?

Response#4

The SiP-Br particles were redispersed in n-octyl acrylate (OA) as a monomer for the ATRP synthesis, as mentioned in Lines 115-116, Page 3.

Comment #5

Line 132 – It is not correct to say the molecular weight of the grafted POA chains were determined by size-exclusion chromatography. Rather, the molecular weight of free solution POA chains grown at the same time as the grafted chains was measured, and it is assumed the molecular weight of the grafts is the same. This assumption should be noted.

Response#5

Based on your suggestion, we have revised the sentences in Lines 132-136, Page 4 of this revised manuscript. In addition, this analytic methodology has been described in the previous report by Ohno and coworkers (Ref. 15). Therefore, we have cited it in the revised manuscript as follows.

The number average molecular weight (Mn) and its distribution (Mw/Mn) of POA chains which were initiated from EBIB grown at the same time as the grafted chains were determined by size-exclusion chromatography (SEC, HLC-8220GPC, TOSOH, Tokyo, Ja-pan) measurements assuming that the values of Mn and Mw/Mn are nearly equal to those of grafted on particles [15].

Comment #6

Line 146 – What is the apparatus used for hot-pressing by hand? Is it possible to add a schematic or photograph of this to the Supplementary Material file?

Response#6

First, the dried SiP-POA was sandwiched between glass substrates. After that, the glass substrate was pressed vertically with tweezers on a hot plate at 100 ºC. The photograph of hot-pressing by hand has been added in Figure S2 in the Supplementary Materials.

Comment #7

Table 2 – What are the units of Mn?

Response#7

In general, the number average molecular weight (Mn) has no units because the molecular weight is defined as the ratio to the molecular weight of a carbon atom.

Comment #8

Line 158 –Note again that the Mn and Mw/Mn values for the grafted chains is assumed to be the same as the free solution polymer.

Response#8

Based on your suggestion, we have revised the sentences in Lines 158-161, Page 4 of this revised manuscript. As related to Comment #5, the previous report by Ohno and coworkers has been referred as Ref. 15 in this sentence.

The Mn, Mw/Mn and graft density (σ) of SiP-POAs with different molecular weights of POA chains are compiled in Table 2. The Mn and Mw/Mn values for the grafted chains are assumed to be the same as the POA chains initiated from EBIB according to the previous report [15].

Comment #9

Line 160 – Units of Mn?

Response#9

Same as Comment #7, the number average molecular weight (Mn) has generally no units.

Comment #10

Line 164 – Is the density of the silica particles, 2.19 g/cm3, assumed from literature, supplied by the manufacturer or directly measured by the authors? If it is measured, either by the manufacturer or authors, please detail how. If it is an assumed value from literature, this is not adequate. It is well-known that the density of nano- and micro-size silica can vary significantly depending on porosity. A measured value must be used.

Response#10

The density of the silica particles (ρ) was taken from the data sheet released in the website of Nippon Shokubai Co., Ltd. In the previously submitted manuscript, the ρ value of “2.19 g/cm3” was typo. Therefore, we have corrected the ρ value as “2.20 g/cm3” according to the data sheet.

Therefore, we have added the following sentence in Lines 171-172, Page 5 of this revised manuscript.

In this study, we adopted the ρ value of 2.20 g/cm3 from the data sheet of Nippon Shokubai Co., Ltd.

Comment #11

Line 189 – “casted” should be “cast”.

Response#11

We appreciate your kind advice. We have revised the sentences in Lines 196-198, Page 5 of this revised manuscript as follows. In addition, we have corrected from “drop-casted” to “cast” in Line 13, Page 1.

As the toluene suspensions of SiP-POAs were cast and hot-pressed according to the procedure mentioned in the Experimental Section, the solid-state CC films of SiP-POAs could be obtained without any covalent crosslinking.

Comment #12

Line 199 – units for molecular weight.

Response#12

Same as Comment #7, the number average molecular weight (Mn) has usually no units.

Round 2

Reviewer 3 Report

The authors have addressed all my concerns, except for providing direct evidence that the packing of the particles changes with the molecular weight of the pol(n-octyl acylate) coating. Relying on the previous work of Ohno and Mizuta is not, in my opinion, sufficient to support the hypothesis since, as note by the authors, the glass transition temperature of POA and PMMA are very different.

I will leave the final decision on this point to the Editor.

Author Response

Manuscript ID: polymers-1895318

Manuscript Title:

Colloidal Crystal Films with Narrow Reflection Bands by Hot-Pressing of Polymer-Grafted Silica Particles

(Previous MS title: Colloidal Crystal Formation of Polymer-Grafted Silica Particles Previous Manuscript ID: polymers-1810530)

Authors:

Sawa Matsuura, Mami Obara, Naoto Iwata* and Seiichi Furumi*

Point-by-point responses. Our responses and opinions are shown in BLUE.

============================================================

To the Reviewer #3

Thank you very much for your second peer-review process on our manuscript.

============================================================

Comment #1

The authors have addressed all my concerns, except for providing direct evidence that the packing of the particles changes with the molecular weight of the pol(n-octyl acylate) coating. Relying on the previous work of Ohno and Mizuta is not, in my opinion, sufficient to support the hypothesis since, as note by the authors, the glass transition temperature of POA and PMMA are very different.

I will leave the final decision on this point to the Editor.

Our Response#1

We have received also the Academic Editor’s comments as follows.

-------------------------------------

I carefully read the text of the resubmitted article and the contents of the reviews, notably the responses from authors to reviewer 3's comments.

I think that doing SEM observations on films is a requirement to the publication of this manuscript.

regards

-------------------------------------

After receiving your and Academic Editor’s comments, we have much effort to observe our colloidal crystal films by SEM. It was not straightforward to observe the microscopic state of particle arrangement in our colloidal crystal films. This is because the top surfaces of our colloidal crystal films are relatively rough. Therefore, we had a hard time to focus the electron beam of SEM on the microscopic area of our colloidal crystal films in order to clearly observe the particle arrangement at a microscopic level.

After our many trials and errors in both sample preparation and SEM observation, we successfully obtained the SEM images of our colloidal crystal films. The SEM images suggest that our polymer-grafted silica particles are packed to form colloidal crystal structures. Moreover, we found that the neighboring distance between polymer-grafted silica particles in our colloidal crystal films extends by increasing the molecular weight of surface-grafted polymer chains.

Taking these experimental results, we have added the results and discussion in this revised manuscript as follows. The SEM images were newly added in Figure 3B of this revised manuscript. The discussion has been mentioned between Line 240, Page 6 and Line 248, Page 7.

Following the new addition of SEM images in Figure 3B, the relationship between the degrees of polymerization (m) of POA chains on the surface of silica particles and the maximum wavelengths of reflection bands (λ) of the colloidal crystals has been moved from Figure 3B (previous manuscript) to Figure 4 (this revised manuscript).

Taking our overall corrections in account, we strongly hope that you find our appropriate revisions of this re-submitted manuscript according to your and the Academic Editor’s comments. Finally, we believe that this revised manuscript deserves urgent publication in “Polymers” as an “Article” in the Special Issue of “Polymer-SiO2 Composites (Edited by Prof. H. M. Cho)”.

============================================================

To the Academic Editor

Thank you very much for your kind comments on our manuscript as follows.

-------------------------------------

I carefully read the text of the resubmitted article and the contents of the reviews, notably the responses from authors to reviewer 3's comments.

I think that doing SEM observations on films is a requirement to the publication of this manuscript.

regards

-------------------------------------

============================================================

Our Response

We have much effort to observe our colloidal crystal films by SEM. It was not straightforward to observe the microscopic state of particle arrangement in our colloidal crystal films. After our many trials and errors in both sample preparation and SEM observation, we successfully obtained the SEM images of our colloidal crystal films. The SEM images suggest that our polymer-grafted silica particles are packed to form colloidal crystals. Moreover, we found that the neighboring distance between polymer-grafted silica particles in our colloidal crystal films extends by increasing the molecular weight of surface-grafted polymer chains.

Taking these experimental results, we have added the results and discussion in this revised manuscript as follows. The SEM images were newly added in Figure 3B of this revised manuscript. The discussion has been mentioned between Line 240, Page 6 and Line 248, Page 7.

Taking our overall corrections in account, we strongly hope that you find our appropriate revisions of this re-submitted manuscript according to your and the Reviewer #3 comments. Finally, we believe that this revised manuscript deserves urgent publication in “Polymers” as an “Article” in the Special Issue of “Polymer-SiO2 Composites (Edited by Prof. H. M. Cho)”.
